# 14,15-EET Reduced Brain Injury from Cerebral Ischemia and Reperfusion via Suppressing Neuronal Parthanatos

**DOI:** 10.3390/ijms22189660

**Published:** 2021-09-07

**Authors:** Haipeng Zhao, Jing Tang, Hongyang Chen, Wei Gu, Huixia Geng, Lai Wang, Yanming Wang

**Affiliations:** 1School of Life Sciences, Henan University, Kaifeng 475000, China; 340809@henu.edu.cn (H.Z.); Tangjing@henu.edu.cn (J.T.); 104753190796@henu.edu.cn (H.C.); Guwei@henu.edu.cn (W.G.); 2Institute of Chronic Disease Risks Assessment, School of Nursing and Health Sciences, Henan University, Kaifeng 475000, China; genghuixia@henu.edu.cn

**Keywords:** 14,15-epoxyeicosatrienoic acids, cerebral ischemia and reperfusion, parthanatos

## Abstract

To investigate the effect of 14,15-EET on the parthanatos in neurons induced by cerebral ischemia and reperfusion, middle cerebral artery occlusion and reperfusion (MCAO/R) and oxygen glucose deprivation/reoxygenation (OGD/R) were used to simulate cerebral ischemia reperfusion in vivo and in vitro, respectively. TTC staining and the Tunel method were used to detect cerebral infarct volume and neuronal apoptosis. Western blot and immunofluorescence were used to detect poly (ADP-ribose) polymerase-1 (PARP-1) activation and AIF nuclear translocation. The production of reactive oxygen species (ROS) and the expression of antioxidant genes were detected by Mito SOX, DCFH-DA and qPCR methods. MCAO/R increased cerebral infarct volume and neuronal apoptosis in mice, while 14,15-EET pretreatment increased cerebral infarct volume and neuronal apoptosis. OGD/R induced reactive oxygen species generation, PARP-1 cleavage, and AIF nuclear translocation in cortical neurons. 14,15-EET pretreatment could enhance the antioxidant gene expression of glutathione peroxidase (GSH-Px), heme oxygenase-1 (HO-1) and superoxide dismutase (SOD) in cortical neurons after ischemia and reperfusion. 14,15-EET inhibits the neuronal parthanatos induced by MCAO/R through upregulation of the expression of antioxidant genes and by reducing the generation of reactive oxygen species. This study advances the EET neuroprotection theory and provides a scientific basis for targeted clinical drugs that reduce neuronal parthanatos following cerebral ischemia and reperfusion.

## 1. Introduction

Ischemic stroke is currently the disease with the highest mortality and disability rates in humans [1,2,3]. There are about 5.5 million annual deaths caused by the disease worldwide [4]. Various injury mechanisms are involved in neuronal apoptosis and neurological damage after stroke. Parthanatos is a unique form of cell death that is mediated by poly ADP-ribose polymerase (PARP) and is completely different from programmed cell apoptosis. It is widely involved in cell oxidative damage and the pathological processes of neurodegenerative disease [5,6,7]. The parthanatos inhibitor is neuroprotective against ischemic stroke that works by suppressing neuroinflammation and matrix metaloproteinase-9 expression and reducing blood–brain barrier damage [8].

PARP is a superfamily of DNA repair enzymes found in eukaryotes. The expression of PARP-1 accounts for more than 85% of all PARP family members [9]. In the case of DNA damage, PARP-1 is activated by DNA double-strand breaks (DSBs) and is cleaved into p89 and p24. The latter irreversibly binds to broken ends of DNA, inhibits DNA repair and leads to cell apoptosis. Activated PARP induces large poly (ADP-ribose) (PAR) polymer formation in cells, and these polymers translocate to the outer mitochondrial membrane to bind the apoptosis-inducing factor (AIF), promoting AIF translocation to the cell nucleus and initiating parthanatos.

Epoxyeicosatrienoic acids (EETs), the metabolites of arachidonic acid, have many neuroprotective effects. EETs can regulate neuronal excitability, increase cerebral blood flow (CBF), inhibit inflammation and neuronal apoptosis, and promote functional recovery in the central nervous system [10]. EETs have four isomers, including 5,6-, 8,9-, 11,12-, and 14,15-EET, which are easily metabolized by soluble epoxide hydrolase (sEH) into dihydroxyeicosatr-ieonic acids (DHETs) with low biological activity. Due to the absence of sEH, the concentration of 14,15-DHET in the brain tissue of sEH gene knockout mice is much lower than that of wildtype mice. Reported results indicate that sEH gene knockout inhibits the metabolism of 14,15-EET and increases the concentration of 14,15-EET correspondingly [11,12].

Our previous studies have showed that sEH gene knockout or 14,15-EET preconditioning can reduce cortical neuronal apoptosis and cerebral infarction volume induced by cerebral ischemia and reperfusion [13,14]. In this study, mice and primary cortical neurons were used as the research objects. The effects of 14,15-EET on neuronal parthanatos induced by cerebral ischemia and reperfusion were investigated through middle cerebral artery occlusion reperfusion and oxygen glucose deprivation/reoxygenation models. This research provides a scientific reference for the in-depth understanding of the neuroprotective mechanism of EET and the development of neuroprotective drugs for cerebral ischemia and reperfusion.

## 2. Results

### 2.1. 14,15-EET Reduces Infarct Volume and Cell Death Induced by Cerebral Ischemia and Reperfusion

TTC staining results showed that cerebral infarct volume increased after MCAO/R. Pretreatment of 14,15-EET with tail vein injection reduced cerebral infarct volume during reperfusion, while injection of PBS had no corresponding protective effect (Figure 1). The mNSS scores of these mice were increased after MCAO/R, yet were reduced with pretreatment of 14,15-EET (Figure 1). Tunel results showed that the cell apoptosis rate was increased in the MCAO/R group, while it was reversed in the pretreatment group with 14,15-EET (Figure 2).

### 2.2. 14,15-EET Inhibits Parthanatos Induced by Cerebral Ischemia and Reperfusion

Both MCAO/R and OGD/R induced the activation of PARP-1 and the increase in poly (ADP-ribosylation) in neurons and led to nuclear translocation of AIF, while 14,15-EET pre-treatment inhibited the activation of PARP-1 and reduced the nuclear translocation of AIF (Figure 3, Figure 4 and Figure 5), suggesting that 14,15-EET pre-treatment can inhibit neuronal parthanatos induced by cerebral ischemia and reperfusion (Figure 3).

### 2.3. 14,15-EET Reduced the Generation of ROS Induced by OGD/R

The results of reactive oxygen species in cortical neurons in vitro showed that OGD/R induced an increase in the generation of reactive oxygen species in neurons, and 14,15-EET pretreatment alleviated this process, while PBS had no protective effect (Figure 6A,B). Furthermore, the results of MitoSOX™ Red staining of mitochondrial reactive oxygen species showed that OGD/R induced an increase in reactive oxygen species in mitochondria, while 14,15-EET treatment reduced ROS (Figure 6C).

### 2.4. 14,15-EET UpRegulated the Transcription of Antioxidant Genes in Neurons after MCAO/R

qPCR results showed that the pretreatment of 14,15-EET via tail vein injection increased the transcription of certain antioxidant genes such as glutathione peroxidase (GSH- Px), heme oxygenase-1 (HO-1) and superoxide dismutase (SOD), while catalase (CAT) gene expression showed no increase, suggesting that 14,15-EET can inhibit the production of reactive oxygen species induced by cerebral ischemia and reperfusion by upregulating the expression of antioxidant genes such as GSH-Px, HO-1 and SOD (Figure 7).

## 3. Discussion

The excessive activation of PARP-1 catalyzes the formation of PAR polymers and induces AIF to translocate from mitochondria to the nucleus, causing large DNA fragmentation and leading to cell death, which is the main process of parthanatos. This does not depend on the activation of caspase and does not result in the formation of apoptotic bodies and small fragments of DNA. It is a special form of “cell death” different from apoptosis. Parthanatos is involved in neuronal loss and nerve function damage in many neurodegenerative diseases, such as Alzheimer’s disease (AD), amyotrophic lateral sclerosis (ALS), Huntington’s disease (HD) and Parkinson’s disease (PD) [5,15]. Therefore, some PARP-1 inhibitors were considered as drug candidates for neurodegenerative diseases such as AD [16,17]. A body of research has demonstrated that parthanatos is also responsible for the pathogenesis of cerebral ischemia and reperfusion. The process of parthanatos inhibition by inhibitors or genetic deletion reduces neuronal cell death and infarct volume, suppresses neuroinflammation, and enhances recovery of neurological functions in cerebral ischemia and reperfusion [8,18,19].

Epoxyeicosatrienoic acid (EET) is the product of the arachidonic acid metabolism through the cytochrome P450 (CYP450) epoxygenase pathway. According to the position of the epoxide ring on EETs, there are four regioisomers, namely, 5,6-EET, 8,9-EET, 11,12-EET and 14,15-EET. Studies have shown that knockout of sEH genes leads to a decrease in 14,15-DHET, the metabolite of EETs, suggesting that knockout of sEH inhibits the metabolism of 14,15-EET and relatively increases the concentration of 14,15-EET in the brain [20]. EETs have many neuroprotective effects, such as regulating nerve excitability, increasing cerebral blood flow, inhibiting inflammation and neuronal apoptosis, and promoting nerve recovery [10]. Our study showed that MCAO/R induces PARP-1 to be cleaved and activated in mouse brains, leading to AIF translocation from mitochondria to the nucleus. 14,15-EET pretreatment can inhibit PARP-1 activation and AIF nuclear translocation and reduce neuronal parthanatos [21].

The generation of reactive oxygen species (ROS), mitochondrial dysfunction and Ca^2+^ overload caused by cerebral ischemia and reperfusion are the main mechanisms of neuronal apoptosis and nerve function injury [22,23]. The inhibition of ROS reduces cell death in hippocampal neurons induced by oxygen and glucose deprivation in vitro [24]. Reactive oxygen species are the main inducers of neuronal parthanatos [21]. A number of studies have shown that parthanatos is one of the significant pathological mechanisms in cerebral ischemia and reperfusion injury [5,8]. This study shows that MCAO/R or OGD/R treatment causes PARP-1 to assume a truncated active form, inducing AIF translocation from mitochondria to the nucleus in cortical neurons, leading to increased neuronal parthanatos. 14,15-EET pretreatment reduces the production of reactive oxygen species and inhibits neuronal parthanatos through enhancing the transcription of antioxidant genes such as GSH-Px, HO-1 and SOD. This is consistent with recent studies showing that EET can promote the transcription and expression of antioxidant enzyme genes to exert a cytoprotective effect [25,26,27]. MCAO/R itself can also induce the upregulation of HO-1 gene expression in neurons, but this upregulation did not reach the level required to inhibit parthanatos induced by MCAO/R, while 14,15-EET pretreatment can significantly increase the expression of HO-1 and cooperates with the high expression of GSH-Px and SOD genes to eliminate the reactive oxygen species induced by MCAO/R to exert a neuroprotective effect. 14,15-EET can enhance the transcription and expression of PGC-1a and promote mitochondrial biogenesis. PGC-1 can increase the expression of HO-1, which is consistent with the results of this study [27,28,29].

This study suggested that 14,15-EET can reduce the production of reactive oxygen species induced by cerebral ischemia and reperfusion via upregulation of the expression of antioxidant genes such as HO-1, GSH-Px and SOD, thereby inhibiting the activation of PARP-1 and nuclear translocation of AIF and alleviating parthanatos of neurons (see illustration in Figure 8).

A body of research has shown that EET could relieve diabetes, hypertension, vascular calcification, cardiovascular disease, lung disease and central post-stroke pain [30,31,32,33]. Therefore, the means of regulating EET metabolism or EET analogs may become effective clinical strategies for the prevention and treatment of various diseases in the future. These results not only theoretically advance the neuroprotection theory of EETs but also provide a scientific reference for the development of drugs based on reducing the death of neurons by parthanatos after cerebral ischemia and reperfusion.

## 4. Materials and Methods

### 4.1. Reagents

High-glucose and glucose-free Dulbecco’s modified eagle medium (DMEM), neurobasal medium, B27, MitoSOX™ Red mitochondrial superoxide indicator, Alexa Fluor 488 and 532 labeled fluorescent secondary antibodies, BCA Protein Assay Kit and other reagents were purchased from Thermo Fisher Scientific Inc (ThermoFisher Scientific, Waltham, MA, USA). Poly-L-lysine hydrobromide (P1274) was purchased from Sigma-Aldrich (St. Louis, MO, USA). The rabbit anti-AIF antibody (#4642) and rabbit anti-PARP antibody (#9542) were purchased from Cell Signaling Technology (Danvers, MA, USA). The mouse anti-MAP2 antibody (ab221693) was purchased from Abcam (Abcam, Cambridge, UK). The mouse anti-β-actin antibody (sc-47778) and mouse anti-pADPr antibody (sc-56198) were purchased from Santa Cruz Biotechnology (Santa Cruz, CA, USA). The RIPA lysate was purchased from Kangwei Century Biotechnology (Cwbio, Beijing, China). The Tunel assay kit (G1501) was purchased from Wuhan Servicebio Company (Wuhan, China). The DCFH-DA kit was purchased from Beyotime (Beijing, China). The MCAO thread plug was purchased from RWD life science Co., Ltd. (Shenzhen, China). Other chemical reagents were analytically pure.

### 4.2. Middle Cerebral Artery Occlusion and Reperfusion (MCAO/R)

Male C57BL/6J mice (22–25 g) were purchased from Charles River (Beijing, China). Mice were housed in a specific pathogen-free (SPF) animal facility with constant temperature and humidity and a 12 h light/dark cycle, and they had free access to water or food. All procedures were approved by the Ethics Committee of Henan University School of Medicine. The process of the MCAO/R model was as we previously described with minor modification [13]. The mice were anesthetized with sodium pentobarbital (1%) by intraperitoneal injection. The neck was then depilated, the neck skin cut with surgical scissors and the muscle layer separated under a stereo microscope. The internal carotid (ICA), external carotid artery (ECA) and common carotid arteries (CCA) were exposed and ligated with sutures. A filament was inserted into the ECA and fixed with a slip knot. The slip knot on the ICA was untied. The thread plug was turned homeopathically until its top plugged into the internal carotid artery. In the same direction, the filament was slowly pushed into the upper left and the thread was stopped when a slight resistance was felt. At the same time, the slipknot on the ICA was slightly tied to fix the filament plug, which blocked the middle cerebral artery. Cerebral blood flow (CBF) was measured with a laser Doppler blood perfusion monitor (PeriFlux 5000, Perimed, Sweden). A reduction of 80% in CBF was considered to constitute the success of the MCAO model. The mouse was placed on a constant temperature insulation table (37 °C) to maintain body temperature after the operation. After 120 min, the filament was pulled out and the slipknot on the ICA was fastened to prevent bleeding, which is the reperfusion procedure. Before reperfusion, 14,15-EET (100 nM) or PBS was injected into the tail vein for the EET group and PBS group. Meanwhile, the slipknot on the CAA was untied. The MCAO model was completed after suturing the muscle layer and skin layer. The sham group was subjected to the same steps as the MCAO group except for the use of a filament plug and reperfusion.

### 4.3. Behavioral Assessment

The Modified Neurology Severity Score (mNSS) was used to assess neurobehavior of mice at 22 h after middle cerebral artery occlusion and reperfusion. The mNSS evaluates defects in the motor and sensory function and balance of mice. The process was performed as described with minor modifications [33]. The investigator was blinded to groups when giving a score to every mouse. The higher the score, the more severe the mouse’s neurological defects.

### 4.4. Primary Cortical Neurons Culture

The cell culture plates were coated with 0.5% polylysine at 37 °C overnight. The next day the plates were rinsed with ultra-pure water 3 times and dried on an ultra-clean table for later use. The brains of E16-day fetal mice were taken out quickly, and the left and right cerebral cortexes were peeled off and separated from the connective tissues, such as meninges and blood vessels, under a stereoscopic microscope. They were then cut into small pieces of about 1 mm^3^ with ophthalmic scissors. The tissues were digested with 0.25% trypsin (1:250) solution in a 37 °C water bath for 15 min, the solution gently swirled and mixed every 3 min. The digestion process was terminated by DMEM medium containing 10% FBS. Deoxyribonuclease (DNase 100 U/mL) was added to the tissue mix fluid and blown carefully to a single cell suspension by glass pipette. After filtration with a 200-mesh stainless steel screen, the filtrate was centrifuged at 1000 r/min for 5 min, and the supernatant was discarded. The cells were resuspended in neuron culture seeding solution, counted, and seeded in cell culture plates with a density of 1 × 10^5^ cells/mL. The cells were cultured in an incubator at 37 °C, 95% humidity and 5% CO_2_. At 4 h post-seeding, the culture medium was replaced with neuron maintenance medium and half the medium was changed every 3 days. The ratio of neurons and astrocytes was as high as 98 to 2 in the primary cultures cell at 7 days (Appendix A).

### 4.5. TTC Staining

The mice were anesthetized by an intraperitoneal injection of sodium pentobarbital. The mouse brains were taken out and placed in the mouse brain matrices, frozen at −20 °C for 5 min, and cut into slices 2 mm in thickness. The brain slices were quickly placed in 0.5% 2,3,5-triphenyltetrazolium chloride (TTC) dye solution and then transferred to a 37 °C incubator for 10 min. The slices were fixed with 4% PFA and photographed with a digital camera. The infarct volume was analyzed and calculated with Image J software. Calculation formula: infarct volume = (volume of unlesioned hemisphere—uninfarct volume of lesioned hemisphere)/volume of unlesioned hemisphere × 100%.

### 4.6. Oxygen Glucose Deprivation/Reoxygenation (OGD/R)

OGD/R provides a cerebral I/R injury model in vitro, as has previously been described in detail [13]. Before the OGD/R experiment, the cells were rinsed twice with D-Hanks solution; then, glucose-free DMEM was added to the medium, and the oxygen in the anaerobic tank was replaced with a mixture of 95% N_2_ and 5% CO_2_. The cultures were placed in a 37 °C incubator for 2 h for oxygen and glucose deprivation treatment. The control group was replaced with a high-glucose DMEM medium containing 10% fetal bovine serum and cultured under normal culture conditions at 37 °C and 5% CO_2_ for 2 h. After the OGD, the control group and OGD group cells were taken out at the same time, and the cell culture medium was replaced with neurobasal medium. The neurobasal media containing 20 nM 14,15-EET and PBS were named the OGD/ R + EET group and OGD/R + PBS group, respectively. The cells were cultured in 95% humidified air and 5% CO_2_ at 37 °C for 24 h for reoxygenation and glucose restoration.

### 4.7. Western Blot

The total protein (30 mg) was separated by 10% SDS-PAGE gel electrophoresis with 100 V for 60 min and transferred into a PVDF membrane by the wet method. The membrane was placed in 5% skimmed milk to block for 60 min and incubate the primary antibody overnight at 4 °C (PARP antibody, 1:1000 dilution; pADPr antibody, 1:1000 dilution; β-actin antibody, 1:2000 dilution). On the second day, the membrane was washed with TBST 3 times, 10 min each time; the secondary antibody was incubated for 90 min at room temperature; and the membrane was washed 3 times with TBST for 10 min each time. The ECL supersensitive luminescence solution was used to prepare chemiluminescence.

### 4.8. Immunofluorescence

The brain was cut into 5 μm slices with paraffin microtome, and tissue antigens were retrieved using a microwave. Absorbent paper was used to wipe the water around the slices and an immunohistochemistry pen to circle the tissue before primary antibody (rabbit anti-AIF antibody (1:200); mouse anti-MAP2 antibody (1:300)) was added and the tissues were placed in a wet box overnight at 4 °C. The next day, the wet box was taken out and placed at room temperature for 30 min. The slices were washed with PBS 3 times for 10 min each time. Excess liquid was wiped off, but the tissue pieces were kept moist. Diluted fluorescent secondary antibody (1:500) was added in the dark, followed by incubation at room temperature for 3 h. The tissues were washed with PBS in the dark 3 times, 10 min each time, and then mounted with DAPI glycerol. The images were taken with a fluorescence microscope (Olympus, BX61, Tokyo, Japan).

### 4.9. Measurement of ROS

The measurement of ROS in cortical neurons used a reactive oxygen species assay kit. The neurobasal medium was discarded, and the cell was washed with pre-warmed sterile PBS 3 times. The neurons were incubated with neurobasal medium containing 10 nM DCFH-DA for 20 min at 37 °C in the dark and then washed 3 times with pre-warmed sterile PBS. The images of ROS were captured with a fluorescence microscope with an equal exposure setting in each group. The fluorescent intensity was analyzed using Image pro-Plus software in every group and the production of ROS was indicated. MitoSOX Red was used to measure mitochondrial reactive oxygen species according to the manufacturer’s instructions. The cells were incubated with 5 m MitoSOX Red for 10 min at 37 °C and washed with warm PBS 3 times. The cells were analyzed using the PE channel of the flow cytometer (CytoFLEX, Beckman).

### 4.10. Apoptosis Detection

Cell apoptosis was detected using a Tunel kit in strict accordance with the manufacturer’s instructions. The sections were fixed with 4% paraformaldehyde for 15 min, placed in a fume hood for 15 s and then washed with PBS 3 times for 5 min each time. Proteinase K (20 μg/mL) was then added for 10 min at room temperature. We added 50 μL equilibration buffer dropwise to cover the tissues at room temperature for 20 min. FITC-12 dUTP Labeling Mix was thawed and recombinant TdT enzyme placed on ice. A TdT incubation buffer was then prepared at a ratio of recombinant TdT enzyme: FITC-12 dUTP Labeling Mix: equilibration buffer = 1 μL:5 μL:50 μL. We then added 56 μL TdT incubation buffer to the slices in the wet box to cover the tissues, which were incubated for 120 min at 37 °C. The TdT incubation buffer was discarded, and the tissues were washed 3 times with PBS for 5 min each time. The images were captured by fluorescence microscope with equal exposure setting.

### 4.11. Real-Time PCR (qPCR)

The total RNA from the brain tissue samples was extracted with the TRlzol method according to the manufacturer’s instructions. The 1 mL TRlzol reagent was added to 100 mg tissue. This was then homogenized in a tissue homogenizer (4 °C 60 HZ, 3 min), after which the homogenized sample was left at room temperature for 5 min. We added 253μL of chloroform, and the samples were quickly inverted 5–6 times before being put at room temperature for 2 min and then centrifuged at 12,000× *g* for 5 min at 4 °C in a high-speed low-temperature centrifuge. The upper water phase was extracted to a new EP tube, mixed with 527μL isopropanol for 5 min at room temperature and then centrifuged at 12,000× *g* for 10 min at 4 °C. The supernatant was discarded and the precipitate retained. This was mixed slowly with 1 mL of 75% ethanol, centrifuged at 7500× *g* for 5 min and dried in a fume hood, and an appropriate amount of preheated DEPC-H_2_O was added to dissolve the RNA. RNA concentration was measured with a Nanodrop and stored in an ultra-low temperature refrigerator. The RNA sample was diluted to a suitable concentration, and the reverse transcription system was set (25 °C, 5 min; 42 °C, 30 min; 72 °C, 5 s). The cDNA sample was stored in the refrigerator at −20 °C. SYBR Green was added to set up the qPCR reaction system (94 °C, 3 min; 60 °C, 20 s; 72 °C, 20 s), and the primers were as follows: SOD (F: 5- ACATAGCTTTGCTCCTGCTTG -3; R: 5- ATCATCGGACAGGCCCTACC -3), GSH-Px (F: 5-TGGTCACAGTTTCACAGTACAC-3; R: 5- CCCAGAGGCACCATTACCTT-3), CAT (F: 5′-CAGGAAGGCTTGCTCAGGAA- 3′; R: 5′-AGGACGGGTAATTGCCATTG-3′) and HO-1 (F: 5′-GGGTCCTCACACTCAGTTT-3′; R: 5′-CCAGGCATCTCCCTTCCATTC-3′). Quantitative calculation and analysis of the reaction results are represented by the following equation:
2^−ΔΔCt^ = 2^−(Ct1−Ct2)^ = 2^−^^〔(a−b)−(c−d)^^〕^(1)

### 4.12. Statistics and Data Analysis

GraphPad Prism 8.0 software was used for statistical analysis of the data. All data results are displayed in the form of mean ± standard deviation (mean ± SD). One-way analysis of variance (one-way ANOVA) or an independent t-test was used to compare the means between groups, and *p* < 0.05 was considered statistically significant.

## Figures and Tables

**Figure 1 ijms-22-09660-f001:**
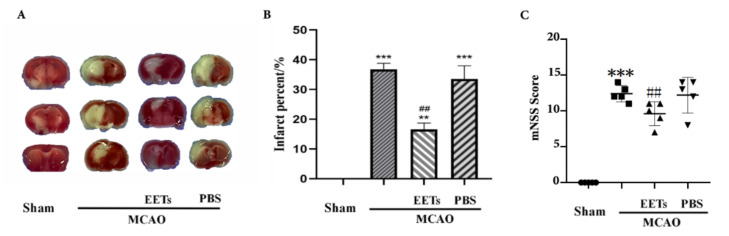
The cerebral infarct volumes were detected by TTC staining method after MCAO and reperfusion. (**A**) TTC staining was used in mouse brain tissue sections at 22 h after middle cerebral artery occlusion and reperfusion. The infarct area appears white. (**B**) Analysis of volume percentage of cerebral infarction (*n* = 3). (**C**) The mNSS score of the mouse at 22 h after middle cerebral artery occlusion and reperfusion (*n* = 5). (*** p* <0.01, **** p* <0.001 vs. sham; *^##^ p* <0.01 vs. MCAO).

**Figure 2 ijms-22-09660-f002:**
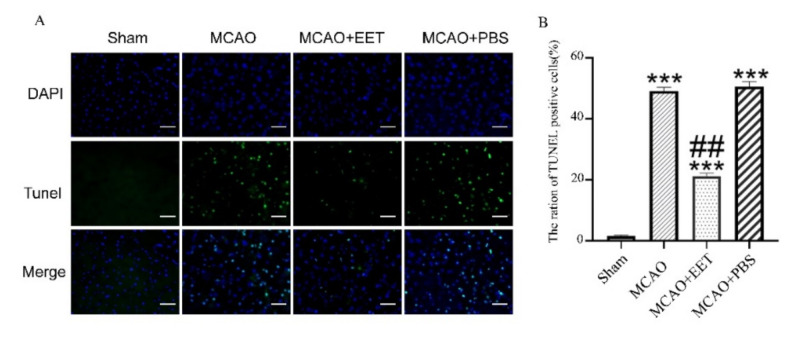
Neuronal apoptosis was detected by Tunel method after MCAO and reperfusion. (**A**) Tunel was used to detect cell death at 22 h after middle cerebral artery occlusion and reperfusion in mice. (**B**) Statistical analysis of apoptosis. (*** *p* < 0.001 vs. sham, ^##^
*p* < 0.01 vs. MCAO; *n* = 3).

**Figure 3 ijms-22-09660-f003:**
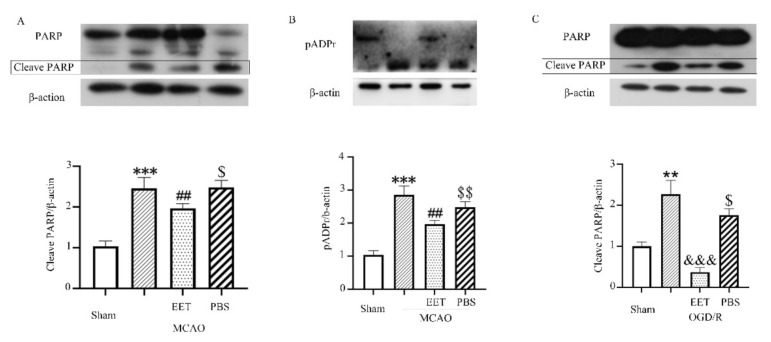
PARP-1 activity following cleavage was detected with Western blot. (**A**) PARP-1 cleavage was detected in total brain protein at 22 h after middle cerebral artery occlusion and reperfusion in mice. (**B**) Poly (ADP-ribosylation) was detected in total brain protein at 22 h after middle cerebral artery occlusion and reperfusion in mice. (**C**) PARP-1 cleavage was detected in total protein of primary cortical neurons at 24 h after OGD/R. (** *p* < 0.01,*** *p* < 0.001 vs. sham; ^##^
*p* < 0.001 vs. MCAO; ^$^
*p* < 0.05, ^$$^
*p* < 0.01 vs. sham; ^&&&^
*p* < 0.001 vs. OGD/R; *n* = 3).

**Figure 4 ijms-22-09660-f004:**
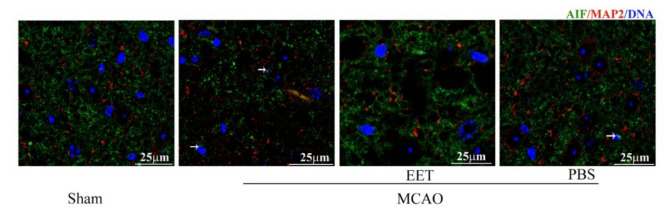
AIF nuclear translocation was detected in mouse brains with immunofluorescence method in cerebral cortical neurons induced at 22 h after MCAO and reperfusion. Arrows point to the nuclear translocation of AIF. 14,15-EET pretreatment reduced the nuclear translocation of AIF at 22 h after middle cerebral artery occlusion and reperfusion in mice.

**Figure 5 ijms-22-09660-f005:**
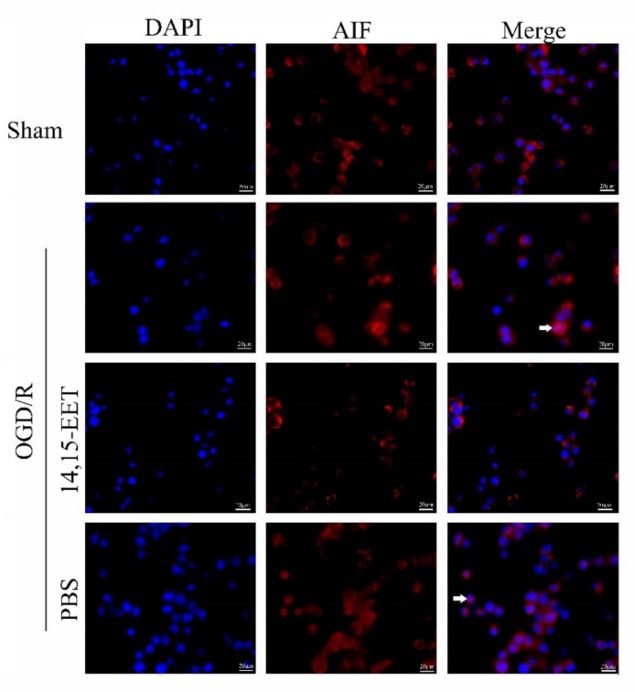
AIF nuclear translocation was detected with immunofluorescence method in primary cortical neurons at 24 h after OGD/R. Arrows point to the nuclear translocation of AIF. 14,15-EET pretreatment reduced the nuclear translocation of AIF in primary cortical neurons at 24 h after OGD/R.

**Figure 6 ijms-22-09660-f006:**
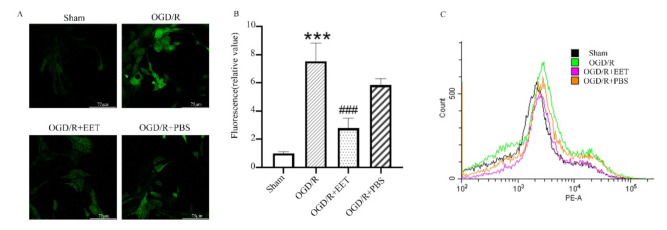
14,15-EET reduced ROS production induced by OGD/R in cortical neurons. (**A**) The generation of ROS was measured with DCFH-DA by confocal laser scanning microscopy in primary cortical neurons at 24 h after OGD/R. (**B**) Statistical analysis of fluorescence intensity of ROS in primary cortical neurons at 24 h after OGD/R. (**C**) Mitochondrial reactive oxygen species were measured with MitoSOX Red staining by flow cytometry. (*** *p* < 0.001 vs. sham; ^###^
*p* < 0.001 vs. OGD/R; *n* = 3).

**Figure 7 ijms-22-09660-f007:**
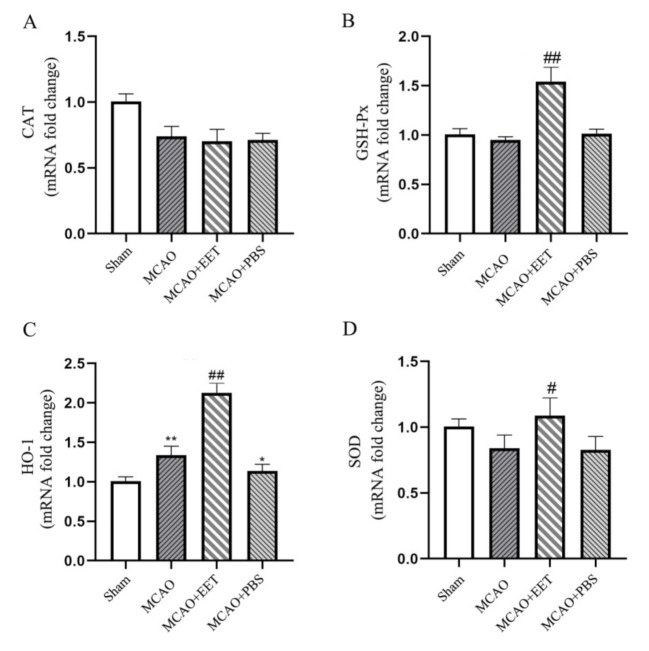
The antioxidant gene expression was detected by qPCR method in brain tissue at 22 h after middle cerebral artery occlusion and reperfusion. (**A**) Catalase. (**B**) Glutathione peroxidase. (**C**) Heme oxygenase-1. (**D**) Superoxide dismutase. (* *p* < 0.05, ** *p* < 0.01 vs. sham; ^#^
*p* < 0.05, ^##^
*p* < 0.01 vs. MCAO; *n* = 3).

**Figure 8 ijms-22-09660-f008:**
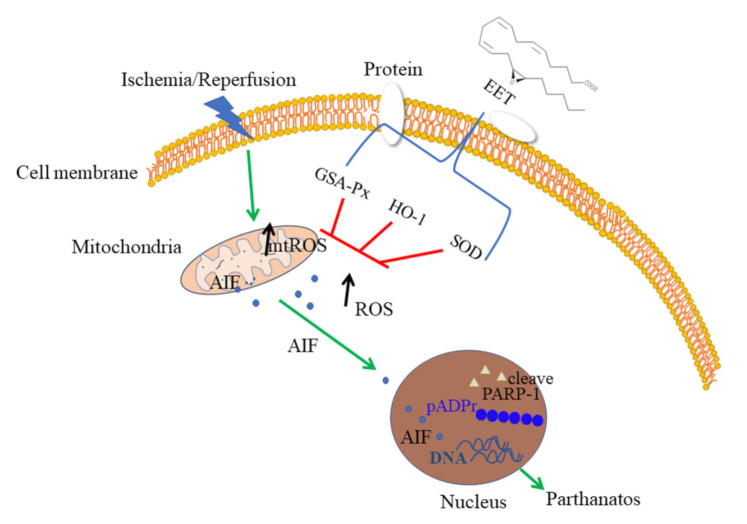
The scheme of neuroprotective function of 14,15-EET. The parthanatos of neurons was caused by the reactive oxygen species being induced by cerebral ischemia and reperfusion. 14,15-EET pretreatment can upregulate the expression of antioxidant genes such as HO-1, GSH-Px and SOD to eliminate reactive oxygen species and inhibit the activation of PARP-1 and nuclear translocation of AIF, reducing neuronal parthanatos.

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
