# Peer review of "14,15-EET Reduced Brain Injury from Cerebral Ischemia and Reperfusion via Suppressing Neuronal Parthanatos"

_ijms, 2021, doi:10.3390/ijms22189660_

Round 1

Reviewer 1 Report

The manuscript described the effects of 14,15-EET on neuronal parthanatos induced by cerebral ischemia and reperfusion. The authors revealed that 14,15-EET pretreatment inhibited such parthanatos. Thus, these findings will be useful for the treatment of cerebral ischemia and reperfusion. Therefore, the manuscript is not too excellent to be published. In other words, the manuscript is so excellent that it should be published.

Comments

(1) How did 14,15-EET reduce the production of ROS induced by OGD/R? In what process, did 14,15-EET upregulate the expression of antioxidant proteins such as GSH-Px, HO-1 and SOD?

(2) Does parthanatos apoptosis occur in chronic brain disease such as Alzheimer’s disease?

(3) Can the epoxy group of 14,15-EET react with proteins in living body, resulting in side effects?

(4) What does 14,15-EET stand for?

(5) In line of 12, “vivo and in vitro respectively” is preferable to be “vivo and in vitro, respectively”.

(6) In line of 30, “annual mortality rate” means not the number but rate.

(7) In line of 65, 78, and 95, “14,15-. EET” is preferable to be “14,15-EET”.

(8) In line of 98, “this process , while” is preferable to be “this process, while”.

That is all.

Reviewer 2 Report

Elucidating the mechanism of neuroprotection by CYP450 eicosanoids and uncovering novel therapeutic applications of these metabolites is an active area of research. The potential role of 14,15-EET in parthanatos is therefore of scientific interest. The authors' main claim is that 14,15-EET inhibits parthanatos in ischemia or glucose/oxygen deprivation by reducing ROS. The results of both in vitro and in vivo models used in this study substantiate the authors' claims. However, the authors' do not provide sufficient data to suggest that the cytoprotection by 14,15-EET is by inhibiting parthanatos as opposed to inhibiting apoptosis or necrosis.  Also, the use of metabolically labile 14,15-EET in the MCAO/R is questionable. Other questions and comments:

The time length for the MCAO (120 min) seems long. The authors' should indicate why this time period was used.

Higher resolution images of Fig. 2 would be nice.

Does 14,15-EET only inhibit parthanatos caused by ROS, or does it also inhibit parthanatos caused by other agents? In vitro data that shows 14,15-EET effects on parthanatos caused by N-methyl-N'-nitrosoguanidine (MNNG) could clarify this. 

The authors' should comment on how they chose the concentration of EET used in the study.  Concentration dependent effects of EET on parthanatos would be a useful piece of data.

The use of other pharmacological tools (e.g., sEH inhibitors) could strengthen the authors' data set.

Data showing PAR accumulation in the models is missing.

In Fig. 6, at what time during OGD/R was ROS production measured?

EETs have four regioisomers or positional isomers (line 49).

Please provide a reference for the statement that 14,15-EET regioisomer has the highest affinity for sEH (line 52).

This is a special form of "cell death" different from apoptosis (line 122).

EETs are the product (not intermediate) of arachidonic acid metabolism through the CYP450 epoxygenase pathway (line 123).

The Discussion section can be improved.

The authors' should include relevant references of 14,15-EET effects on antioxidant gene expression.

Reviewer 3 Report

In this preclinical study, the authors investigated the effect of 14,15-EET on the neuronal parthanatos in the setting of focal brain ischemia/reperfusion in vivo and in vitro. They found that 14,15-EET pretreatment reduced neuronal apoptosis after ischemia and reperfusion, accompanying with higher expression of GSH-Px, HO-1 and SOD in cortical neurons and less PARP-1 activation/AIF nuclear translocation. In conclusion, 14,15-EET inhibits the neuronal parthanatos induced by I/R through up-regulating the expression of antioxidant genes and reducing the generation of reactive oxygen species.

  1. The study is descriptive and lacking the mechanistic experiments to support the cause-effect of anti-oxidant effects of EET and neuronal parthanatos suppression.
  2. The clinical relevance of pretreatment approach is not well justified.
  3. TUNEL staining is not specific to apoptosis. 
  4. Sample size n=3 is relative low for statistical analysis in animal study. 5
  5. There is no evaluation of neurobehavioral outcome. 
  6. Fig 4 and 5 were used to demonstrate the AIF nuclear translocation. It is not clear what the arrows indicates and what the group difference is.
  7. In Fig. 3A, the internal control bands of beta-actin did not show a comparable density among all groups, which raised the concern of protein loading consistency. Also, there are multiple bands shown for Cleave PARP in A. Which bands were used for quantitative analysis?

Reviewer 4 Report

The article presents new data that will be of interest to a wide range of specialists in the field of neurobiology. The article is well written. Nevertheless, I have a number of suggestions for authors which can improve the article.

  1. Figure legends should be more informative
  2. The text of the article should clearly indicate the ratio of neurons and astrocytes in the cultures of the cerebral cortex.
  3. Can the authors measure ROS production by mitochondria?
  4. The authors do not discuss the role of Ca2 + ions in the regulation of cell death. Ref: 

    https://pubmed.ncbi.nlm.nih.gov/32219700/

    https://pubmed.ncbi.nlm.nih.gov/30776416/

  5. The putative pattern of neuroprotective action of 14,15-EET would be of interest. I suggest including the theoretical scheme of neuroprotective action of  14,15-EET at the end of the article

Best regards,

Round 2

Reviewer 2 Report

Thank you for making significant effort to improve the manuscript. Minor changes:

Line 17: MCAO/R increased (not induced) cerebral infarct volume

Line 45-46: which promotes (not prometing) AIF translocation (not translocate) to the cell nucleus (not nuclear)

Line 50-51: EETs have four regioisomers, 5,6-, 8,9-, 11,12-, and 14,15-EET,

Line 53: Due to the absence of sEH (not Due to the difference in molecular structure),

Line 158: of the epoxide (not oxidation) ring on EETs,

Reviewer 3 Report

The authors have addressed most of the comments in the revision. A few concern remain:

  1. The pretreatment approach is not translatable to the clinical setting  unless the authors suggest the 14,15-EET can be used in stroke patients prophylactically. If so, the relevant discussion is necessary.
  2. If arrows point to the nuclear translocation of AIF in Figs 4&5, a higher magnification of representative microphotographs are needed to show this phenomenon.     

Reviewer 4 Report

The authors took into account all the comments. The manuscript has been significantly revised. I recommend that the authors carefully reread the manuscript again for minor errors. Good luck to the authors.
